# A New Macrodiolide and Two New Polycyclic Chromones from the Fungus *Penicillium* sp. SCSIO041218

**DOI:** 10.3390/molecules24091686

**Published:** 2019-04-30

**Authors:** Jingxia Huang, Jianglian She, Xiliang Yang, Juan Liu, Xuefeng Zhou, Bin Yang

**Affiliations:** 1Zhongshan Ophthalmic Center, Sun Yat-Sen University, Guangzhou 510060, China; 13694217880@163.com; 2Department of Pharmacy, Hubei Province Key Laboratory of Occupational Hazard Identification and Control, Institute of Infection, Immunology and Tumor Microenvironments, Medical College, Wuhan University of Science of Technology, Wuhan 430081, China; sjlsjl0210@163.com; 3CAS Key Laboratory of Tropical Marine Bio-resources and Ecology/Guangdong Key Laboratory of Marine Materia Medica, South China Sea Institute of Oceanology, Chinese Academy of Sciences, Guangzhou 510301, China; liujuan@scsio.ac.cn (J.L.); xfzhou@scsio.ac.cn (X.Z.);

**Keywords:** marine fungus, *Penicillium* sp., penixanthone, macrodiolide, chromone

## Abstract

A new macrodiolide, mangrovlide A (**1**) and two new polycyclic chromones, penixanthones C (**2**) and D (**3**), as well as four other known compounds (**4**–**7**), have been isolated from the mangrove sediment derived fungus *Penicillium* sp. SCSIO041218, cultured in the 0.25% NaCl rice substrate. The structures of the new compounds were determined by analysis of the NMR and MS spectroscopic data. Compound **1** possesses a 10-membered macrodiolide unit, while **2** and **3** are chromones with an unprecedented 6/6/6/5 polycyclic skeleton. Compounds **1**–**7** were evaluated for their cytotoxicities, while all the compounds displayed weak or no activity.

## 1. Introduction

Fungi are remarkable organisms that have a capacity to produce diverse classes of secondary metabolites with a wide variety of biological activities, such as antimicrobial, anti-HIV, antitumor, anti-inflammatory, and enzymatic inhibitory properties [1,2,3,4]. The genus *Penicillium*, widely found in nature, is among the most studied fungi and represents important drug producers [5,6]. One strain-many compounds (OSMAC) strategy was employed in our investigation on the secondary metabolites of *Penicillium* sp. SCSIO041218 in various media. Our previous studies on this fungus, cultured in the 0.25% NaCl rice substrate, have resulted in discovery of three new diprenylated indole alkaloids, mangrovamides A–C [7]. However, these alkaloids failed to appear in 3% NaCl rice substrate, which two new xanthone derivatives, penixanthones A and B, were produced [8]. In our continuing search in this productive strain, four new prenylated indole alkaloids and four new chromone derivatives have been isolated in the 1% NaCl potato dextrose broth (PDB) medium [9]. The present investigation on the other fractions of the 0.25% NaCl rice substrate led to the isolation of a new macrodiolide, mangrovlide A (**1**) and two new polycyclic chromones, penixanthones C (**2**) and D (**3**), as well as four other known compounds 4-hydroxy-6-methoxy-5-methylphthalid (**4**), methyl 2-(5-hydroxy-2,3,4-trimethylphenyl) propionate (**5**), dehydromevalonic lactone (**6**), (*R*)-mevalonolactone (**7**) (Figure 1). In this paper, we report the isolation, structure elucidation, and the bioactivities of these compounds.

## 2. Results

Compound **1** was obtained as a colorless oil. Its molecular formula was assigned as C_14_H_20_O_8_ based on the positive HR-ESI-MS at *m*/*z* 339.1059 [M + Na]^+^, accounting for five degrees of unsaturation. Analysis of ^13^C NMR (Table 1, Appendix A) and distortionless enhancement by polarization transfer (DEPTs) spectra revealed the presence of four carbonyl carbons, two methines, six methylenes (including two O-methylenes), and two O-methyl. Comprehensive analysis of the 2D NMR spectra of **1**, especially ^1^H–^1^H homonuclear correlated spectroscopy (COSY) and heteronuclear multiple bond correlation (HMBC), allowed the establishment of the structure. Two structural fragments [C-2/C-3/C-4/C-5, C-2′/C-3′(C-6′)/C-4′)] were established by the correlations observed in the ^1^H–^1^H COSY spectrum (Figure 2). This structure was also supported by the correlations of H-2 to C-1, C-3, and C-6, H-5 to C-4, and C-3, H-2′ to C-1′, H-4′ to C-2′, C-3′, C-5′, and C-6′, H-6′ to C-2′, and C-3′ in the HMBC experiment. The four carbonyl carbons accounted for four out of five degrees of unsaturation, and the remaining one degree of unsaturation required the presence of one additional ring in **1**. Taking the molecular formula into account, the planar structure of **1**, consisting of 10-membered macrodiolide unit, could then be constructed as shown in Figure 1. The configurations of **1** were not determined, because extensive efforts were unsuccessful, including produce suitable crystals for X-ray diffraction. Thus, the planar structure was established and assigned the trivial name mangrovlide A (**1**).

Compound **2** was obtained as colorless syrup. Its molecular formula was established as C_20_H_20_O_7_ on the basis of the positive HRESIMS at *m*/*z* 373.1276 [M + H]^+^, 395.1092 [M + Na]^+^, accounting for eleven degrees of unsaturation. The ^1^H NMR spectrum (Table 2, Appendix A) showed three aromatic and one olefinic protons [*δ*_H_ 7.39 (1H, t, *J* = 8.5 Hz, H-3), 6.59 (1H, d, *J* = 8.5 Hz, H-4), 6.44 (1H, d, *J* = 8.5 Hz, H-2), 6.03 (1H, s, H-10)], two methyl singlets [*δ*_H_ 2.02 (3H, s, H-17), 1.38 (3H, s, H-18)], one methyl triplet [*δ*_H_ 1.29 (3H, t, *J* = 7.5 Hz, H-20)], and one *O*-methylene [*δ*_H_ 4.26 (2H, m, H-19)]. The ^13^C NMR, DEPT, and heteronculear single quantum coherence (HSQC) data revealed the presence of three methyls (*δ*_C_ 13.4, 14.2, 19.2), two sp^3^ methylenes (*δ*_C_ 32.0, 61.7), one sp^3^ methine (*δ*_C_ 52.5), three sp^3^ quaternary carbons (*δ*_C_ 61.2, 86.1, 92.4), four sp^2^ methines (*δ*_C_ 107.9, 111.3, 127.0, 138.5), four sp^2^ quaternary carbon (*δ*_C_ 105.8, 156.9, 160.1, 163.4), one carbonyl carbon (*δ*_C_ 170.8) and two ketones (*δ*_C_ 191.0, 193.6). Since eight of the eleven degrees of unsaturation were attributed to one aromatic moiety, one trisubstituted double bond, one carbonyl and two ketones, **2** was assumed to contain three other rings.

Through careful analysis of the 1D and 2D NMR data, **2** was found to be similar to mangrovamide J [9], which was isolated from the same fungus with 1% NaCl PDB substrate. HMBC correlations from H-2 to C-1, and C-6; H-3 to C-1, and C-5; and H-4 to C-5 led to the connectivity of the subunits to form a trisubstituted aromatic moiety. Also, the HMBC correlations from H-10 to C-12, C-13 and C-17; H_3_-17 to C-10, C-11, and C-12; and H_3_-18 to C-8, C-12, and C-13, constructed the typical chromone three rings in **2**. The remaining signals in **2** were completely different from other chromone derivatives. Since the preceding data accounted for 11 degrees of unsaturation, the presence of an additional ring was required to satisfy the molecular formula of **2**. Key HMBC correlations from H-14 to C-7, C-8, C-9, C-12, C-13, C-15 and C-16; H-15 to C-11, C-12, C-14, and C-16 revealed the connectivity between C-8 and C-13. Furthermore, one ethyl group positioned at C-19 was confirmed from the HMBC correlations from *δ*_H_ 4.26 (H-19) to the ester carbonyl carbon resonating at *δ*_C_ 170.8. Thus, the gross structure of penixanthone C was assigned as **2**. In comparison to the well-established chromone derivatives, **2** possesses an unprecedented signature C_2_ bridge and corresponding fused five member ring [10,11]. In the nuclear overhauser effect spectroscopy (NOESY) spectrum (Figure 3), correlations of H_3_-17 to H-10 and H-14*β* to H-15 were found. Due to the baseline ECD curves and barely measurable specific rotation values, compound **2** was presumed to be racemic. 

Compound **3** was obtained as colorless syrup. Its molecular formula was established as C_19_H_18_O_7_ on the basis of the positive HRESIMS at *m*/*z* 359.1127 [M + H]^+^, 381.0941 [M + Na]^+^, accounting for eleven degrees of unsaturation. The ^1^H and ^13^C NMR spectra (Table 2, Appendix A) of **3** were similar to those of **2**, with the exception that the ethyl group of the latter was replaced by a methyl singlet at *δ* 3.76. So, the structure of **3** was established and assigned the trivial name penixanthone D (**3**), as shown in Figure 1.

The identities of compounds **4**–**7** were established by comparison of their spectral data with those of the known compounds reported. They are 4-hydroxy-6-methoxy-5-methylphthalid (**4**) [12,13,14], methyl 2-(5-hydroxy-2,3,4-trimethylphenyl) propionate (**5**) [15], dehydromevalonic lactone (**6**) [16], (*R*)-mevalonolactone (**7**) [17,18].

Compounds **1**–**7** were tested for their in vitro cytotoxicities against 10 human tumor cell lines (H1975, U937, K562, BGC823, MOLT-4, MCF-7, A549, Hela, HL60 and Huh-7). Only **2** and **3** showed weak activities against K562 (**2**: 55.2 µM, **3**: 56.5 µM), MCF-7 (**2**: 61.1 µM, **3**: 58.6 µM), Huh-7 (**2**: 67.5 µM, **3**: 64.2 µM) cell lines, with IC_50_ values of 55.2–67.5 µM. Trichostatin A was used as the positive control with IC_50_ values of 0.027 µM–0.162 µM.

## 3. Experimental Section 

### 3.1. General Experimental Procedures

The NMR spectra were recorded on a Bruker AC 500 NMR spectrometer (BrukerBioSpin, Fällanden, Switzerland) with TMS as an internal standard. ESI-MS data were measured on a BrukeramaZon SL spectrometer (Bruker, Fällanden, Switzerland). HR-ESI-MS data were measured on a Bruker micro TOF-QII mass spectrometer (Bruker, Fällanden, Switzerland). UV spectrum was measured by using a Shimadzu UV-2600 PC spectrometer (Shimadzu, Beijing, China). IR spectrum was recorded on an IR-Affinity-1 spectrometer (Shimadzu, Beijing, China). CD spectra were measured with Chirascan circular dichroism spectrometer (Applied Photophysics Ltd., Leatherhead, UK). Optical rotation values were measured with an Anton Paar MCP500 polarimeter (Anton Paar, Graz, Austria). YMC gel (ODS-A, 12 nm, S-50 mm, YMC, Kyoto, Japan) was used for column chromatography. Phenomenex column (Lux Cellulose-2 column, 4.6 mm × 25 mm, Phenomenex, Torrance, CA, USA) was used for chiral analysis HPLC chromatography. The SiO_2_ GF_254_ used for TLC was supplied by the Qingdao Marine Chemical Factory, Qingdao, China. Sephadex LH-20 gel (GE Healthcare, Uppsala, Sweden) was used. HPLC was carried out on Hitachi L-2400 with YMC ODS column (Hitachi, Tokyo, Japan). Spots were detected on TLC under UV light or by heating after spraying with 5% H_2_SO_4_ in EtOH (*v*/*v*).

### 3.2. Fungal Material

The culture of *Penicillium* sp. SCSIO041218 (Original Number: SYFz-1) was isolated from a mangrove sediment sample collected in Sanya (18°13′50.2″N, 109°37′15.8″E) in August 2010. The strain (accession NO. JN713906.1) was identified as *Penicillium* sp. based on a molecular biological protocol calling for DNA amplification and ITS region sequence comparison with the GenBank database and shared a similarity of 100% with *Penicillium funiculosum*. The strain was deposited in the RNAM Center, South China Sea Institute of Oceanology, Chinese Academy of Sciences, Guangzhou, China.

### 3.3. Fermentation and Purification

Strain stored on PDA slants at 4 °C was cultured on PDA agar plates and incubated for 7 days. Seed medium (potato 200 g, dextrose 20 g, NaCl 2.5 g, distilled water 1000 mL) in 50-mL Erlenmeyer flasks was inoculated with strain F00120 and incubated at 25 °C for 48 h on a rotating shaker (180 rpm). Production medium of solid rice in 1000 mL flasks (rice 200 g, NaCl 0.5 g, distilled water 200 mL) was inoculated with 10 mL seed solution. Flasks were incubated at 25 °C under static stations and daylight. After 50 days, cultures from 10 flasks were harvested for the isolation of substances.

The total 2 kg of rice culture was crushed and extracted with acetone three times. The acetone extract was evaporated under reduced pressure to afford an aqueous solution, and then the aqueous solution was extracted with EtOAc to yield 29 g of a crude gum. The EtOAc portion was subsequently separated by Si gel column chromatography using CHCl_3_–MeOH gradient elution to give seventeen fractions (*Frs. A*~*Q*). *Fr. D* was purified by semipreparative RP HPLC (70% MeOH in H_2_O) at a flow rate of 3 mL/min to afford **2** (3.3 mg), **3** (2.4 mg). *Fr. H* was further separated by semipreparative reversed-phase HPLC (75% MeOH in H_2_O) at a flow rate of 3 mL/min to afford **6** (4.6 mg), and **7** (3.1 mg). *Fr. J* was further purified by CC over ODS with MPLC, using gradient elution from 20% MeOH to 100% at a flow rate of 15 mL/min, to yield 12 portions (*Fr. J*–1~*Fr. J*–12). *Fr. J*–4 was further separated by semipreparative reversed-phase HPLC (55% MeOH in H_2_O) at a flow rate of 3 mL/min to afford **1** (5.6 mg). *Fr. K* was further purified by CC over ODS with MPLC, using gradient elution from 20% MeOH to 100% at a flow rate of 15 mL/min, to yield 17 portions (*Fr. K*–1~*Fr. K*–17). *Fr. J*–5 was further separated by semipreparative reversed-phase HPLC (45% MeOH in H_2_O) at a flow rate of 3 mL/min to afford **5** (3.0 mg). *Fr. M* was further separated by semipreparative reversed-phase HPLC (45% MeOH in H_2_O) at a flow rate of 3 mL/min to afford **4** (4.0 mg).

Mangrovlide A (**1**): colorless oil; ${\left[\alpha \right]}_{D}^{25}$ = −3.1 (0.1, *c*, MeOH); UV (MeOH) λ_max_ (log ε): 204 (0.74) nm; IR (film) *v*_max_: 3365, 1645, 1016 cm^−1^; ^1^H and ^13^C NMR in Table 1; HR-ESI–MS *m*/*z* 339.1059 [M + Na]^+^ (Calcd for C_14_H_20_NaO_8_ 339.1050).

Penixanthone C (**2**): colorless syrup; ${\left[\alpha \right]}_{D}^{25}$ = −10.4 (0.1, *c*, MeOH); UV (MeOH) λ_max_ (log ε): 207 (1.63), 274 (0.61), 353 (0.19) nm; IR (film) *v*_max_: 3481, 1622, 1205, 1010 cm^−1^; ^1^H and ^13^C NMR in Table 2; HR-ESI–MS *m*/*z* 373.1276 [M + H]^+^ (Calcd for C_20_H_21_O_7_ 373.1282), *m*/*z* 395.1092 [M + Na]^+^ (Calcd for C_20_H_20_NaO_7_ 395.1088).

Penixanthone D (**3**): colorless syrup; ${\left[\alpha \right]}_{D}^{25}$ = −6.8 (0.12, *c*, MeOH); UV (MeOH) λ_max_ (log ε): 207 (1.63), 274 (0.61), 353 (0.19) nm; IR (film) *v*_max_: 3361, 1624, 1014 cm^−1^; ^1^H and ^13^C NMR in Table 2; HR-ESI–MS *m*/*z* 359.1127 [M + H]^+^ (Calcd for C_19_H_19_O_7_ 359.1125), *m*/*z* 381.0941 [M + Na]^+^ (Calcd for C_19_H_18_NaO_7_ 381.0945).

## 4. Conclusions

The investigation of bioactive natural products from the fungus *Penicillium* sp. SCSIO041218, cultured in the 0.25% NaCl rice substrate, have resulted in discovery of a new macrodiolide, mangrovlide A (**1**) and two new polycyclic chromones, penixanthones L (**2**) and M (**3**), along with four other known compounds 4-hydroxy-6-methoxy-5-methylphthalid (**4**), methyl 2-(5-hydroxy-2,3,4-trimethylphenyl) propionate (**5**), dehydromevalonic lactone (**6**), and (*R*)-mevalonolactone (**7**). They feature a unique natural product framework. The unique structure and chemical properties of **1**–**3** are attractive for chemists and biologists.

## Figures and Tables

**Figure 1 molecules-24-01686-f001:**
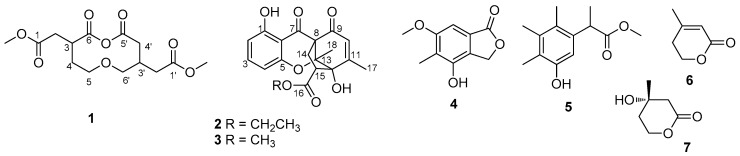
Structures of compounds **1**–**7**.

**Figure 2 molecules-24-01686-f002:**
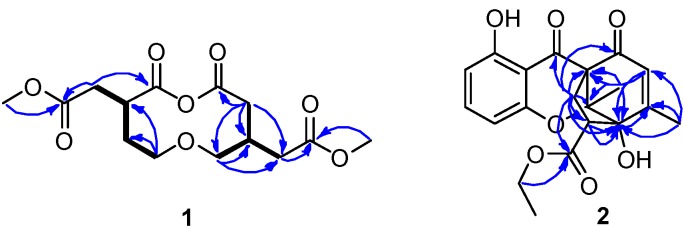
Key HMBC and ^1^H–^1^H COSY correlations of compounds **1** and **2**.

**Figure 3 molecules-24-01686-f003:**
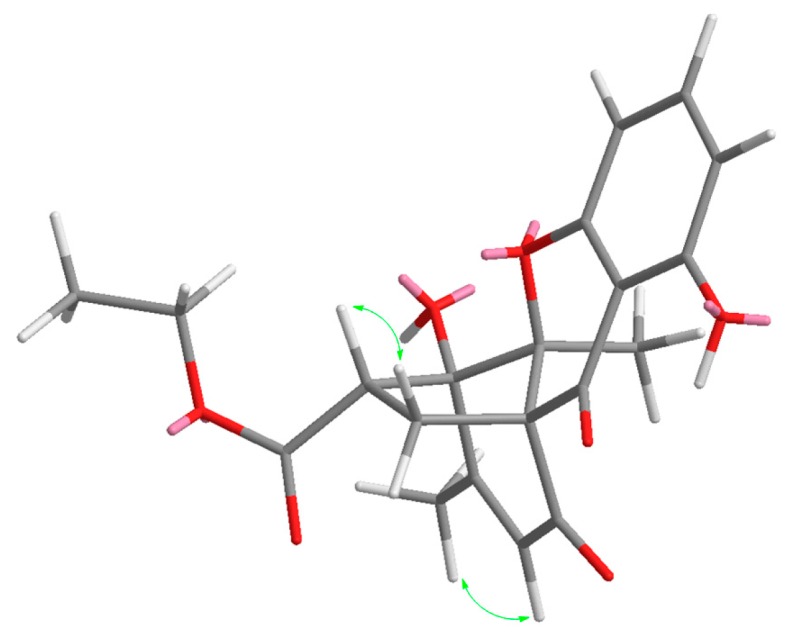
NOESY correlations of compound **2**.

**Table 1 molecules-24-01686-t001:** ^1^H and ^13^C NMR data for **1** (500/125 MHz, in CD_3_OD).

Position	*δ*_H_, mult, *J* in Hz	*δ* _C_
1		172.0
2	2.83 dd (12.5, 3.5)	33.4
	2.67 dd (12.5, 5.5)	
3	3.05 m	35.8
4	2.52 m	27.8
	2.14 m	
5	4.43 td (6.5, 1.5)	66.8
	4.29 ddd (9.0, 5.0, 5.0)	
6		179.6
OMe	3.71 s	50.8
1’		172.2
2’	2.59 dd (5.0, 1.5)	36.1
3’	2.99 dt (11.5, 5.0)	31.8
4’	2.75 dd (12.5, 6.5)	33.3
	2.36 dd (12.5, 5.5)	
5’		178.1
6’	4.55 dd (6.5, 5.5)	72.8
	4.06 dd (6.5, 5.0)	
OMe	3.70 s	50.9

**Table 2 molecules-24-01686-t002:** ^1^H and ^13^C NMR data for **2** and **3** (500/125 MHz, in CDCl_3_).

Position	2	3
*δ*_H_, mult, *J* in Hz	*δ* _C_	*δ*_H_, mult, *J* in Hz	*δ* _C_
1		156.9		156.9
2	6.44 d (8.5)	107.9	6.45 d (8.0)	107.9
3	7.39 t (8.5)	138.5	7.40 t (8.0)	138.5
4	6.59 d (8.5)	111.3	6.60 d (8.0)	111.4
5		163.4		163.4
6		105.8		106.0
7		191.0		191.1
8		61.2		61.2
9		193.6		193.5
10	6.03 s	127.0	6.02 s	127.1
11		160.1		160.0
12		86.1		86.2
13		92.4		92.4
14α	2.59 dd (7.0, 15.0)	32.0	2.60 dd (7.0, 15.0)	32.0
14β	2.46 dd (10.0, 15.0)		2.46 dd (10.0, 14.5)	
15	3.73 m	52.5	3.74 t (8.0)	52.5
16		170.8		171.2
17	2.02 s	19.2	2.00 s	19.0
18	1.38 s	13.4	1.37 s	13.4
19	4.26 m	61.7	3.76 s	52.5
20	1.29 t (7.5)	14.2

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
