# Peer review of "A New Macrodiolide and Two New Polycyclic Chromones from the Fungus *Penicillium* sp. SCSIO041218"

_molecules, 2019, doi:10.3390/molecules24091686_

Round 1

Reviewer 1 Report

Dear, Xiliang Yang, Ph.D and Bin Yang, Ph.D.

Please check.

-------------------------------------------------------------------------------------------------------------------------

Compound 1

Would you detect HMBC correlations H-5 to C5’ and H-6’ to C-6?

-------------------------------------------------------------------------------------------------------------------------

Author Response

COMPOUND 1

Would you detect HMBC correlations H-5 to C5’ and H-6’ to C-6?

Response: Unfortunately, we don’t detect HMBC correlations from H-5 to C-5’ and H-6’ to C-6.

Reviewer 2 Report

MS #: molecules-492510

Title: A New Macrodiolide and Two New Polycyclic Chromones from the Fungus Penicillium sp. SCSIO041218

I had gone through the revised manuscript and I felt that this manuscript was not revised carefully in the present form. Although authors had provided some more important information, they did not organize the manuscript well to meet the criteria of this journal. Therefore, this manuscript is not recommended to accept for publication in Molecules in the present form. It should be reconsidered the acceptance after authors′ careful revisions and resubmission. In addition, there were some major comments addressed as following.

1.          According to the spectra provided in the Supplementary Materials, the purity of compounds 2 and 3 were very bad. Authors declared that the specific rotation values of 2 was barely measured. Maybe it was due to the low purity rather than its racemic condition.

2.          In the experimental section, authors had provided the spectra data for the new compounds. However, some problems were raised. Optical rotation should not be reported in degree. What are the IR bands 2358 or 2341 cm-1 presented for? Where is the ester carbonyl group?

3.          The bioactivity of isolated compounds was not significant. In addition, authors have to provide the detail of cytotoxicity examination at least in the Supplementary Materials.

4.          In the References section, the writing manner of references still did not follow the style of this journal. Authors have to revise these style errors again.

Author Response

1.           According to the spectra provided in the Supplementary Materials, the purity of compounds 2 and 3 were very bad. Authors declared that the specific rotation values of 2 was barely measured. Maybe it was due to the low purity rather than its racemic condition

Response: Compounds 2 and 3 were analysed by HPLC chromatography (Eclipse XDB-C18 column, 5μm, 4.6 mm × 250 mm, Agilent), using two kinds of elution solvents (MeOH/H2O and CH3CN/H2O). They are only one peak. When they were analysed by chiral analysis HPLC column (Phenomenex Lux Cellulose-2 column, 4.6 mm × 25 mm, eluent n-hexane/iso-propanol, 60:40 v/v, 1 mL/min), there are a pair of peaks. I think that the purity of compound 2 is good. Due to the sample shortage, compound 3 was not further purified. However, the NMR spectra of 3 were similar to those of 2. Compound 3 could be easily determined by comparison of NMR spectra.

2.           In the experimental section, authors had provided the spectra data for the new compounds. However, some problems were raised. Optical rotation should not be reported in degree. What are the IR bands 2358 or 2341 cm-1 presented for? Where is the ester carbonyl group?

Response: Have delecteded, please see the text. I had made a mistake. The IR bands at 2358 or 2341 cm-1 are definitely related to CO2, which come from ambient circumstances.

3.           The bioactivity of isolated compounds was not significant. In addition, authors have to provide the detail of cytotoxicity examination at least in the Supplementary Materials.

Response: Have changed, please see the text and the Supplementary Materials.

4.            In the References section, the writing manner of references still did not follow the style of this journal. Authors have to revise these style errors again.

Response: Have changed, please see the text. We have used the EndNote to manage these references, according the style of Molecules.

Round 2

Reviewer 2 Report

I had gone through the revised manuscript carefully and this manuscript was recommended for acceptance after some minor revisions since authors had provided suitable explanation for most of the previous queries. Some minor concerns were addressed as follows.

1. The physical state of compounds 2 and 3 should be revised as “syrup” rather than “oil”.

2. There were still some format errors to be found in the text, such as some italic fonts should be revised as normal fonts. Authors have to check and revise these errors.

3. Lines 182-189, the contents of Supplementary Materials should include biological assays. In addition, I suggested that the descriptions of supp. Figures could be briefed.

Author Response

1.      The physical state of compounds 2 and 3 should be revised as “syrup” rather than “oil”.

Response: Have changed, please see the text

2.      There were still some format errors to be found in the text, such as some italic fonts should be revised as normal fonts. Authors have to check and revise these errors.

Response: Have changed, please see the text

3.      Lines 182-189, the contents of Supplementary Materials should include biological assays. In addition, I suggested that the descriptions of supp. Figures could be briefed.

Response: Have changed, please see the text